# Beyond the Tissue: Unlocking NSCLC Treatment Potential Through Liquid Biopsy

**DOI:** 10.3390/genes16080954

**Published:** 2025-08-13

**Authors:** Milica Kontic, Mihailo Stjepanovic, Filip Markovic

**Affiliations:** 1Clinic for Pulmonology, University Clinical Center of Serbia, Koste Todorovica 26, 11000 Belgrade, Serbia; 2School of Medicine, University of Belgrade, Dr Subotica 8, 11000 Belgrade, Serbia

**Keywords:** non-small-cell lung cancer (NSCLC), liquid biopsy, circulating tumor DNA (ctDNA), immune checkpoint inhibitors (ICIs), tumor mutational burden (TMB), biomarkers, precision oncology

## Abstract

Lung cancer (LC), with non-small-cell lung cancer (NSCLC) as its predominant subtype, remains the leading cause of cancer-related mortality worldwide. While immune checkpoint inhibitors (ICIs) have redefined the therapeutic paradigm in advanced NSCLC, durable responses are confined to a limited subset of patients. A major clinical challenge persists: the inability to accurately predict which patients will derive meaningful benefit, which will exhibit primary resistance, and which are at risk for severe immune-related toxicities. The imperative to individualize ICI therapy necessitates robust, dynamic, and accessible biomarkers. Liquid biopsy has emerged as a transformative, minimally invasive tool that enables real-time molecular and immunologic profiling. Through analysis of circulating tumor DNA (ctDNA), circulating tumor cells (CTCs), exosomes, and peripheral blood immune components, liquid biopsy offers a window into both tumor intrinsic and host-related determinants of ICI response. These biomarkers not only hold promise for identifying predictive signatures—such as tumor mutational burden, neoantigen landscape, or immune activation states—but also for uncovering mechanisms of acquired resistance and guiding treatment adaptation. Beyond immunotherapy, liquid biopsy plays an increasingly central role in the landscape of targeted therapies, allowing early detection of actionable driver mutations and resistance mechanisms (e.g., EGFR T790M, MET amplification, and ALK fusion variants). Importantly, serial sampling via liquid biopsy facilitates longitudinal disease monitoring and timely therapeutic intervention without the need for repeated tissue biopsies. By guiding therapy selection, monitoring response, and detecting resistance early, liquid biopsy has the potential to significantly improve outcomes in NSCLC.

## 1. Introduction

Lung cancer continues to represent the most lethal malignancy globally, with non-small-cell lung cancer (NSCLC) accounting for nearly 85% of all diagnosed cases [1]. Despite advances in systemic therapies, including the introduction of immune checkpoint inhibitors (ICIs) and targeted therapy, treatment resistance and heterogeneous responses remain persistent clinical challenges [2]. The lack of reliable predictive biomarkers hampers the ability to tailor therapy effectively and avoid unnecessary toxicities. In this context, liquid biopsy has emerged as a promising non-invasive approach for the dynamic evaluation of tumor and immune characteristics [3].

A major hurdle in optimizing the use of ICIs lies in the absence of reliable biomarkers capable of accurately predicting which patients will benefit from treatment. Current strategies such as PD-L1 expression on tumor cells or tumor mutational burden (TMB) in tissue biopsies offer limited predictive value and are confounded by spatial and temporal tumor heterogeneity [4]. Moreover, tissue biopsies are invasive, may not be feasible in all patients, and cannot be repeated frequently to monitor disease evolution. As a result, clinicians often lack the tools to dynamically tailor immunotherapy, risking both under- and overtreatment.

Liquid biopsy refers to the sampling and analysis of non-solid biological tissue, primarily blood, to detect tumor-derived components such as circulating tumor DNA (ctDNA), circulating tumor cells (CTCs), exosomes, and other immune-related markers [5]. Unlike traditional tissue biopsy, which provides a static snapshot of the tumor at a specific time and location, liquid biopsy enables serial monitoring of disease evolution, treatment response, and resistance mechanisms. Recent technological advancements have enabled increasingly sensitive and specific detection of tumor- and immune-related biomarkers in the bloodstream. These developments have facilitated novel insights into treatment response, resistance mechanisms, minimal residual disease, and even immune-related adverse events [6,7]. The integration of liquid biopsy into clinical workflows could transform the way we approach immunotherapy and targeted in NSCLC, shifting from reactive to adaptive, biomarker-guided treatment strategies.

This review comprehensively examines the emerging role of liquid biopsy in the context of immunotherapy and targeted for NSCLC. We focus on key analyzes such as ctDNA, mutational profiling of actionable and resistance genes, soluble immune checkpoints (e.g., sPD-L1), peripheral blood immune cell ratios (e.g., NLR), and plasma-based tumor mutational burden (cTMB) (Figure 1). We also address the current limitations of liquid biopsy technologies and propose future directions for integrating these biomarkers into routine clinical decision making.

## 2. Biological Rationale and Technical Advances in Liquid Biopsy

The rationale behind liquid biopsy lies in its ability to non-invasively capture tumor heterogeneity and molecular evolution. ctDNA, a fragment of DNA shed into the bloodstream from apoptotic or necrotic tumor cells, reflects tumor-specific genetic alterations and is increasingly recognized as a surrogate for tumor burden. Recent advancements in next-generation sequencing (NGS), digital PCR, and ultra-sensitive droplet digital PCR (ddPCR) technologies have significantly improved the detection limits and specificity of ctDNA assays.

In addition to ctDNA, circulating tumor cells (CTCs), extracellular vesicles (EVs), and soluble immune checkpoint molecules provide complementary information regarding tumor biology, immune interactions, and microenvironmental context. Each of these analytes offers unique advantages and limitations, and their combined evaluation may improve the sensitivity and clinical utility of liquid biopsy as a comprehensive biomarker platform.

## 3. ctDNA Kinetics as a Predictor of Immunotherapy and Targeted Therapy Response

This section focuses on circulating tumor DNA (ctDNA), a cell-free nucleic acid biomarker, and its role in predicting treatment response, monitoring minimal residual disease, and detecting molecular resistance during immunotherapy and targeted therapy in NSCLC.

The dynamic changes in ctDNA levels during ICI treatment have emerged as valuable predictors of treatment efficacy. Early decreases in ctDNA levels after the initiation of therapy have been associated with favorable clinical outcomes, including longer progression-free survival (PFS) and overall survival (OS). These observations suggest that ctDNA may serve as an early on-treatment biomarker, potentially preceding radiographic responses and helping to identify responders and non-responders more accurately [8].

In a multicenter prospective study evaluating circulating tumor DNA (ctDNA) dynamics in patients with advanced NSCLC treated with pembrolizumab, early reduction in ctDNA levels—particularly within the first two cycles of therapy—was strongly associated with improved clinical outcomes. Specifically, patients who exhibited a ≥50% decrease in ctDNA variant allele frequency (VAF) demonstrated significantly prolonged progression-free survival (PFS) and overall survival (OS) compared to those with stable or rising ctDNA levels [9]. These findings underscore the potential of ctDNA as a real-time, non-invasive biomarker for on-treatment response monitoring.

This concept was reinforced in exploratory analyses from the IMpower150 trial, which evaluated atezolizumab in combination with bevacizumab and chemotherapy in patients with metastatic nonsquamous NSCLC. In this setting, ctDNA clearance—defined as undetectable levels of tumor-specific mutations in plasma—was associated with superior survival outcomes. Patients with ctDNA clearance at early on-treatment timepoints had a median OS of 25.5 months compared to 13.4 months in those without clearance [10]. These findings support the use of ctDNA kinetics not only as a prognostic tool, but potentially as a predictive biomarker for treatment efficacy [6]. Beyond response monitoring, ctDNA offers a means to assess minimal residual disease (MRD), a key determinant of long-term outcomes in patients who achieve partial or complete responses. Several studies have shown that the presence of residual ctDNA following initial tumor regression portends a high risk of relapse, even in radiographically stable disease [11]. Conversely, patients who enter molecular remission—defined by undetectable ctDNA at multiple consecutive timepoints—often experience durable benefit, suggesting that ctDNA negativity reflects profound immunologic control over residual tumor cells.

In one longitudinal study by Chaudhuri et al., patients with localized or oligometastatic NSCLC who had undergone curative-intent therapy were monitored using ultra-sensitive ctDNA assays. Among those who remained ctDNA-negative post-treatment, none experienced recurrence within the first year, whereas ctDNA-positive patients had a median lead time of 5.2 months before radiographic progression [11]. These findings have important implications for guiding post-ICI surveillance and adjuvant strategies. Emerging evidence also supports the role of ctDNA in guiding treatment duration. In the CheckMate 153 study, continuous nivolumab therapy was compared to fixed-duration treatment. Although not formally stratified by ctDNA status, patients with prolonged benefit after discontinuation could plausibly have been those who achieved molecular remission, underscoring the need to integrate ctDNA into treatment de-escalation trials [12].

Importantly, ctDNA dynamics may also inform treatment intensification. Rising ctDNA levels during therapy have been linked to molecular progression and resistance, often weeks before clinical or radiographic evidence of relapse. Early identification of such trends could trigger timely interventions, including switching to alternative agents, combining with targeted therapies, or enrolling patients in trials of novel immunotherapeutic strategies [13,14].

In addition to its growing role in monitoring response to immunotherapy, ctDNA dynamics have also been actively investigated in patients receiving targeted therapies. Among these, EGFR-mutant NSCLC represents the most extensively studied subgroup, offering valuable insights into how ctDNA can be used to track treatment response, detect resistance, and inform prognosis in the context of targeted therapy.

Clinical trials such as AURA3 and FLAURA have demonstrated that ctDNA analysis for EGFR mutations can serve as an early predictor of treatment response in patients with EGFR-mutant NSCLC [15,16]. In both trials, ctDNA dynamics assessed as early as three weeks after treatment initiation correlated with clinical outcomes and showed potential as a supplementary tool for early detection of disease progression [17]. Similarly, in the SWOG S1403 trial, serial ctDNA monitoring revealed that complete clearance of EGFR mutations by week 8 was significantly associated with longer PFS and overall survival OS, while the persistence of ctDNA was linked to poorer outcomes [18,19]. These findings underscore the value of ctDNA not only for response monitoring but also for identifying patients at higher risk of early progression.

In addition to its role in monitoring, ctDNA is emerging as a valuable prognostic marker. Data from the FLAURA trial showed that patients who were ctDNA EGFR mutation-negative at baseline had significantly longer PFS in both the osimertinib and comparator EGFR-TKI arms compared to ctDNA-positive patients [20]. Additional ctDNA analysis can detect co-mutations linked to poorer prognosis, enabling more personalized and timely treatment decisions. For instance, co-mutations in genes like TP53, RB1, NF1, ARID1A, BRCA1, and PTEN are associated with worse outcomes in patients with EGFR-mutant NSCLC [21]. This suggests that baseline ctDNA status may reflect underlying tumor burden or disease biology and can help stratify patients by risk. As such, incorporating baseline ctDNA analysis into clinical practice could improve individualized treatment strategies by identifying patients who may benefit from closer monitoring or more aggressive therapy upfront.

An emerging application of liquid biopsy in NSCLC is monitoring dynamic changes in circulating tumor DNA (ctDNA) as a surrogate marker of treatment response. Early decreases in ctDNA levels, often occurring before radiologic changes, have been proposed to complement traditional RECIST criteria—a concept known as liquid biopsy RECIST (LB-RECIST) [22]. While promising, this approach is still in early clinical development and requires further validation in larger trials.

Analytically, the sensitivity of ctDNA assays continues to improve, with technologies such as hybrid-capture NGS, personalized digital PCR, and molecular barcoding enabling detection at variant allele frequencies below 0.1%. However, challenges remain, including the standardization of thresholds for ctDNA clearance, the need for tumor-informed assays in low-shedding tumors, and false negatives due to clonal hematopoiesis [23]. Nonetheless, as ctDNA testing matures, it is poised to become a cornerstone in adaptive, biomarker-guided immuno- and targeted therapy for NSCLC.

## 4. Mutation Profiling via ctDNA

In this section, we examine ctDNA as a tool for comprehensive genomic profiling in NSCLC, emphasizing its utility in detecting predictive and resistance mutations relevant to targeted therapy and immunotherapy.

### 4.1. Predictive and Resistance Biomarkers for Targeted Therapy

Detection of EGFR mutations in peripheral blood using ctDNA is now a well-established tool in routine clinical practice for NSCLC. The Cobas EGFR Mutation Test v2 was the first liquid biopsy approved as a companion diagnostic to identify patients with the EGFR T790M mutation eligible for third-generation EGFR tyrosine kinase inhibitors (TKIs) [24,25]. Initially introduced to the field of NSCLC for identifying EGFR-sensitizing mutations, ctDNA testing has since evolved—particularly with the advent of NGS—to support comprehensive genomic profiling, enabling broader assessment of tumor heterogeneity and resistance mechanisms. In patients receiving targeted therapies, ctDNA analysis allows for early detection of resistance mutations, often preceding radiographic progression [26]. A well-known example is the emergence of the aforementioned EGFR T790M mutation in patients treated with first- or second-generation EGFR TKIs [26]. Moreover, recent prospective real-world data have shown that liquid biopsy significantly shortens turnaround time for EGFR mutation testing (3 vs. 26 days; *p* < 0.001) compared to tissue NGS [27]. Importantly, liquid biopsy enabled 43.7% of patients to initiate targeted therapy before tissue results were available, emphasizing its clinical utility in expediting treatment decisions and improving access to precision therapies [27].

Liquid biopsy has broadened the molecular profiling capabilities in advanced NSCLC by allowing detection of important genetic alterations beyond EGFR mutations. KRAS and BRAF mutations are frequently identified in the blood of patients whose tumors carry these changes, though the predictive and prognostic roles of liquid biopsy for these mutations need further validation [28,29]. Ongoing exploratory studies aim to further clarify and address these challenges [30,31]. Additionally, rare alterations in RET and MET genes have been found in ctDNA from patients with advanced cancers, highlighting the expanding clinical potential of liquid biopsy [32,33,34]. While ctDNA testing for gene fusions such as ALK rearrangements and ROS1 fusions is under active investigation, its clinical use for detection of resistance mutations is limited by lower sensitivity and a higher likelihood of false negatives—especially when the circulating tumor DNA fraction is low [[35],[36],,[37]]. However, when the tumor fraction exceeds 1%, the accuracy of ctDNA assays improves significantly, with concordance rates comparable to tissue-based testing [38]. In these cases, liquid biopsy offers a reliable and less invasive alternative for molecular diagnosis, particularly when tissue samples are insufficient or unavailable, helping to guide timely and personalized treatment decisions. Similar to EGFR-TKIs, resistance to ALK inhibitors typically emerges after a median of 9–10 months, with mutations such as L1196M, G1269A, F1174L/C, C1156Y, and G1202R accounting for approximately 30% of cases [39]. Some of these resistance mutations are targetable with second- and third-generation ALK inhibitors [40]. Liquid biopsy has proven useful in detecting these ALK resistance mutations at the time of disease progression, offering a non-invasive tool for guiding subsequent treatment strategies.

### 4.2. Predictive and Resistance Biomarkers for ICI

Beyond quantifying tumor burden, ctDNA offers a powerful platform for comprehensive genomic profiling of NSCLC, enabling the identification of mutations that may predict differential sensitivity or resistance to immune checkpoint inhibitors (ICIs). Somatic alterations in key tumor suppressors and oncogenes have been shown to influence the tumor microenvironment, modulate immune signaling, and ultimately affect immunotherapy outcomes.

Notably, mutations in STK11 (also known as LKB1) and KEAP1 are recurrent in NSCLC and have been independently associated with primary resistance to PD-1/PD-L1 blockade. STK11 loss impairs AMPK signaling and induces an immune-cold phenotype characterized by reduced CD8+ T-cell infiltration and low PD-L1 expression [41]. KEAP1 mutations disrupt the NRF2 pathway, promoting antioxidant signaling and suppressing pro-inflammatory responses within the tumor niche [42]. Similarly, PTEN loss has been linked to impaired interferon signaling and reduced T-cell infiltration, contributing to ICI resistance in preclinical and clinical settings [43].

In contrast, certain co-mutation patterns appear to confer enhanced immunogenicity. Co-occurring TP53 and KRAS mutations have been associated with increased tumor mutational burden (TMB), elevated neoantigen loads, and a pro-inflammatory immune milieu that favors ICI responsiveness [44]. In multiple retrospective cohorts, these co-mutations have been linked to improved progression-free and overall survival under PD- (L)1 therapy compared to tumors harboring KRAS mutations alone [45,46].

The longitudinal tracking of variant allele frequencies (VAFs) of somatic mutations via ctDNA provides an additional layer of insight. Reductions of >50% in VAF during early treatment cycles have been associated with favorable clinical outcomes and correlate with radiographic tumor regression. In several studies, these molecular shifts preceded imaging-based response assessments by 4–8 weeks, suggesting that ctDNA dynamics could serve as an early on-treatment biomarker [47,48].

Furthermore, emerging evidence suggests that mutations in chromatin remodeling genes, including ARID1A and ARID1B, may sensitize tumors to ICIs. Loss of ARID1A has been associated with impaired mismatch repair and increased TMB, both of which enhance neoantigen presentation and immune recognition [48]. In preclinical models, ARID1A-deficient tumors exhibit heightened interferon signaling and greater infiltration by effector T cells, supporting the notion that these mutations could function as positive predictive biomarkers for immunotherapy [49].

Collectively, these data underscore the potential of ctDNA not only as a biomarker of tumor burden but also as a non-invasive tool for molecular stratification. By enabling real-time assessment of clonal dynamics and mutational evolution, ctDNA can inform precision immunotherapy strategies and support the transition from static to adaptive treatment paradigms in NSCLC (Figure 2).

## 5. Soluble Immune Checkpoint Molecules in Peripheral Blood

This section discusses soluble immune checkpoint proteins, primarily sPD-L1 and sPD-1, as circulating protein biomarkers that may reflect immune activity and predict clinical outcomes during ICI treatment. In addition to tumor-derived genetic material, liquid biopsy platforms can evaluate immune-related soluble proteins in plasma, offering further insight into the host immune response and tumor–immune interaction. Among these, soluble forms of immune checkpoint molecules—particularly soluble programmed death-ligand 1 (sPD-L1) and soluble programmed death-1 (sPD-1)—have gained attention as potential non-invasive biomarkers in NSCLC.

Soluble PD-L1 (sPD-L1) is primarily derived from proteolytic cleavage of membrane-bound PD-L1 or, less frequently, from alternative mRNA splicing variants lacking transmembrane domains [50]. Quantification of sPD-L1 is typically performed using enzyme-linked immunosorbent assays (ELISA), although newer multiplexed immunoassay platforms and electrochemiluminescence methods have been developed for higher sensitivity and specificity [51]. Elevated baseline or on-treatment levels of sPD-L1 have consistently been associated with poor prognosis in NSCLC. In a meta-analysis of 1162 patients across 11 studies, high circulating sPD-L1 levels were significantly correlated with inferior overall survival (OS) and progression-free survival (PFS), irrespective of PD-L1 expression in tumor tissue [52]. These findings suggest that sPD-L1 may reflect systemic immunosuppression or active PD-L1 shedding by tumor and immune cells as a mechanism of immune escape. Interestingly, sPD-L1 levels do not always correlate with tumor PD-L1 expression as assessed by immunohistochemistry, highlighting the possibility that the circulating form may be regulated independently or produced by non-malignant stromal and immune cells within the tumor microenvironment [53]. Some reports also indicate that treatment-induced changes in sPD-L1 during ICI therapy may be more informative than static baseline levels. For instance, rising sPD-L1 concentrations during treatment have been associated with disease progression and inferior clinical outcomes [54]. The biological function of sPD-L1 remains under investigation. Preclinical studies have shown that circulating sPD-L1 retains the ability to bind PD-1 on T cells, leading to functional suppression of T cell activation and cytokine production, potentially recapitulating the immunosuppressive effects of the membrane-bound protein [55].

In contrast, the role of soluble PD-1 (sPD-1) is less clearly defined. Some studies suggest that sPD-1 may act as a decoy receptor, sequestering PD-L1 and thereby preventing inhibitory signaling through the PD-1/PD-L1 axis. In patients receiving ICIs, increasing levels of sPD-1 during therapy have been observed in those achieving durable clinical benefit, possibly reflecting immune activation or T-cell proliferation in response to treatment [56]. However, data remain inconsistent, and the prognostic or predictive value of sPD-1 requires further validation.

Other soluble immune checkpoint molecules—including CTLA-4, TIM-3, LAG-3, and TIGIT—are under active investigation. Elevated plasma levels of soluble CTLA-4 have been linked to poor outcomes in melanoma and lung cancer, though their role in NSCLC immunotherapy remains to be established [57]. Multiplex proteomic profiling of soluble immune regulators may offer additional value in the development of multi-analyte predictive signatures.

Overall, the assessment of circulating immune checkpoint proteins offers a non-invasive means to monitor systemic immune dynamics and immunotherapy response. While promising, these markers currently lack standardized thresholds, and inter-assay variability limits their clinical implementation. Prospective trials with harmonized biomarker assessment protocols are needed to fully elucidate the utility of soluble checkpoint molecules in guiding immunotherapy decisions.

## 6. Inflammatory Biomarkers and Hematologic Indices

Here, we address hematologic indices derived from routine blood tests—such as neutrophil-to-lymphocyte ratio (NLR), platelet-to-lymphocyte ratio (PLR), and systemic immune-inflammation index (SII)—as indirect markers of systemic inflammation and immune status in NSCLC patients undergoing immunotherapy. Systemic inflammation plays a pivotal role in shaping the tumor microenvironment and modulating the efficacy of immune checkpoint inhibitors (ICIs). Among the most accessible biomarkers for evaluating systemic immune status are hematologic indices derived from routine complete blood counts (CBC), including the neutrophil-to-lymphocyte ratio (NLR), platelet-to-lymphocyte ratio (PLR), and systemic immune-inflammation index (SII). These metrics offer cost-effective, universally available surrogates of host immune status and tumor-associated inflammation.

The neutrophil-to-lymphocyte ratio (NLR) has emerged as the most studied inflammatory biomarker in the context of ICI therapy. Elevated baseline NLR (typically >3 or >5, depending on the study) has been consistently associated with worse progression-free survival (PFS) and overall survival (OS) in patients with advanced NSCLC treated with anti–PD-1 or anti–PD-L1 agents [58,59]. Neutrophils contribute to an immunosuppressive milieu through secretion of reactive oxygen species, matrix metalloproteinases, and neutrophil extracellular traps (NETs), which can hinder T-cell infiltration and facilitate tumor progression [60]. Conversely, lymphocytes, particularly CD8+ T cells, are critical for antitumor immunity, and a relative reduction in their numbers may signal impaired immune surveillance. A recent meta-analysis encompassing over 3000 patients with NSCLC receiving ICIs confirmed that high baseline NLR was significantly associated with reduced OS (hazard ratio [HR]: 2.05; 95% CI: 1.76–2.39) and PFS (HR: 1.83; 95% CI: 1.55–2.15) [61]. Furthermore, dynamic changes in NLR during treatment may offer additional prognostic value. Rising NLR within the first 4–8 weeks of therapy has been linked to early disease progression and inferior clinical benefit [62].

Platelet-to-lymphocyte ratio (PLR) and systemic immune-inflammation index (SII)—which incorporates neutrophil, lymphocyte, and platelet counts—have also been evaluated as prognostic indicators. Elevated PLR has been associated with poor survival outcomes, potentially reflecting platelet-mediated tumor protection and angiogenesis [63]. The Systemic Immune-Inflammation Index (SII)—a blood-based marker that reflects the balance between inflammation and immune response in the body—is calculated by multiplying the number of neutrophils and platelets, then dividing by the number of lymphocytes—with higher values indicating a more immunosuppressive and pro-tumor environment. High SII has been independently correlated with reduced response rates and survival in ICI-treated NSCLC cohorts, suggesting a compounded effect of neutrophilia and thrombocytosis in promoting immune evasion [64]. Although these indices lack tumor specificity and are influenced by comorbid conditions (e.g., infections, corticosteroid use, hematologic disorders), their appeal lies in their simplicity, reproducibility, and low cost, making them ideal candidates for inclusion in multi-analyte biomarker models. Emerging studies have also begun exploring integrative scoring systems that combine hematologic indices with molecular or imaging biomarkers. For example, the Lung Immune Prognostic Index (LIPI)—which incorporates derived NLR (dNLR) and lactate dehydrogenase (LDH) levels—has shown promise in stratifying NSCLC patients receiving ICIs into distinct prognostic groups with differing response rates and survival outcomes [65]. Such tools may aid in refining patient selection and optimizing immunotherapy strategies in clinical practice.

## 7. Tumor Mutational Burden (TMB) from ctDNA (cTMB)

Tumor mutational burden (TMB), when assessed via ctDNA (cTMB), represents a genomic biomarker quantifying the mutational load in circulating tumor DNA. This section explores cTMB as a predictor of response to immune checkpoint inhibitors. Tumor mutational burden (TMB) represents the total number of somatic mutations per megabase (mut/Mb) of the tumor genome and serves as a surrogate for neoantigen load, which may promote recognition by the immune system [66]. In the era of immunotherapy, TMB has emerged as a potential predictive biomarker for response to immune checkpoint inhibitors (ICIs) across various malignancies, including non-small-cell lung cancer (NSCLC) [67].

Traditionally, TMB is assessed using formalin-fixed paraffin-embedded (FFPE) tumor tissue via whole-exome sequencing or large targeted next-generation sequencing (NGS) panels (tTMB). However, tissue-based TMB analysis faces limitations, including the need for sufficient tumor material, invasive biopsy procedures, spatial heterogeneity, and inability to perform dynamic monitoring [68]. These constraints have led to increasing interest in plasma-derived TMB (cTMB), calculated from circulating tumor DNA (ctDNA), as a non-invasive alternative that enables real-time evaluation of mutational burden.

Multiple clinical studies have demonstrated that high cTMB is associated with improved responses to ICIs in NSCLC. In the prospective BF1RST trial, patients with advanced NSCLC treated with first-line atezolizumab showed improved outcomes if they had a baseline cTMB ≥ 10 mut/Mb, including higher objective response rates (ORR: 28.6% vs. 4.4%) and prolonged progression-free survival (PFS) [69]. Similarly, data from the B-F1RST update and POPLAR/OAK studies have supported cTMB as a prognostic and predictive marker in both first-line and previously treated settings [70].

Despite these encouraging findings, discrepancies between cTMB and tissue-based TMB are common and arise due to several factors: variations in assay sensitivity and panel size, differences in ctDNA shedding (especially in low-burden or indolent tumors), and technical challenges related to background noise and clonal hematopoiesis [71,72]. While tTMB typically captures a broader mutational landscape, cTMB may reflect a more current and systemic picture of tumor burden.

Recent work has focused on harmonizing cTMB thresholds and standardizing assay platforms. In 2022, an international consensus proposed technical guidelines for cTMB measurement, recommending minimum panel sizes of 1.5–2.0 Mb and rigorous bioinformatic filtering to exclude germline and hematopoietic variants [73]. Efforts to validate concordance between commercial assays (e.g., GuardantOMNI, FoundationOne Liquid CDx, Tempus xF) are ongoing, with promising but variable results.

Importantly, cTMB alone may not fully capture the complexity of ICI response. As such, several groups have proposed composite biomarker models that integrate cTMB with PD-L1 expression, ctDNA dynamics, immune cell infiltration, and soluble immune checkpoint levels [74,75]. For example, in a recent 2023 study by Yang et al., combining cTMB with dynamic ctDNA response and plasma PD-L1 levels significantly improved the predictive accuracy for ICI benefit in NSCLC patients, outperforming single-biomarker models [76].

Going forward, the utility of cTMB may lie in its role as part of a multidimensional biomarker panel, guiding not only patient selection for ICIs but also treatment sequencing, escalation, and surveillance strategies. Prospective validation in randomized clinical trials and regulatory harmonization remain essential to facilitate broader clinical adoption.

## 8. Clinical Utility and Implementation Challenges

Despite the growing body of evidence supporting the utility of liquid biopsy in guiding immunotherapy for non-small-cell lung cancer (NSCLC), its widespread clinical adoption remains hindered by several practical and regulatory barriers. While liquid biopsy offers a compelling alternative to tissue sampling—particularly in patients with limited biopsy access, poor performance status, or advanced disease—it is not yet uniformly integrated into clinical workflows.

One of the most significant challenges is analytical variability among liquid biopsy platforms. Differences in assay design, gene panel size, sequencing depth, bioinformatic pipelines, and variant calling criteria can lead to substantial inconsistencies in detecting mutations, calculating tumor mutational burden (cTMB), or quantifying ctDNA dynamics [77]. A study comparing commercial ctDNA assays in lung cancer found only moderate concordance in detecting actionable mutations and cTMB thresholds, highlighting the need for harmonization [78].

Pre-analytical factors, including sample collection methods (e.g., choice of tubes, time to processing), plasma isolation protocols, and ctDNA extraction efficiency, can introduce variability that affects test reliability. Delays in sample processing or improper storage conditions may lead to nucleic acid degradation or leukocyte lysis, increasing background noise and diluting the tumor signal [79]. Thus, standardized operating procedures (SOPs) across laboratories are essential to ensure data reproducibility and clinical trustworthiness.

From a regulatory perspective, the use of liquid biopsy in guiding immunotherapy decisions remains mostly confined to investigational or laboratory-developed tests. Although some ctDNA-based assays (e.g., Guardant360 CDx, FoundationOne Liquid CDx) have received FDA approval for detecting targetable mutations, their utility in immunotherapy prediction—such as for cTMB or minimal residual disease (MRD)—is still undergoing prospective validation [80]. Additionally, reimbursement for liquid biopsy remains inconsistent across healthcare systems, particularly for serial monitoring, limiting its accessibility in routine care.

Clinical interpretation of liquid biopsy results presents another layer of complexity. For example, distinguishing between clonal hematopoiesis-derived mutations and true tumor-derived variants requires careful bioinformatic filtering and, ideally, matched white blood cell sequencing. Moreover, dynamic ctDNA changes, such as transient spikes during pseudoprogression, must be interpreted cautiously in the context of radiologic and clinical data to avoid premature therapy discontinuation [81].

Despite these challenges, the trajectory toward clinical implementation is promising. Multiple large-scale trials are currently evaluating the role of ctDNA-guided treatment adaptation in NSCLC. Liquid biopsy’s key advantages—real-time monitoring, minimal invasiveness, and capacity to reflect spatial and temporal tumor heterogeneity—make it uniquely suited for precision immunotherapy. With the ongoing development of multi-modal biomarker models that combine ctDNA, plasma proteomics, and immune profiling, liquid biopsy is poised to become a cornerstone of personalized oncology.

## 9. Future Directions and Conclusions

The role of liquid biopsy in the management of non-small-cell lung cancer (NSCLC) is undergoing a paradigm shift, transitioning from a complementary diagnostic tool to a central component of precision oncology. As immunotherapy and targeted treatment regimens become increasingly complex and personalized, there is a growing need for dynamic, minimally invasive, and multifactorial biomarkers capable of capturing real-time tumor behavior and host immune dynamics.

The integration of ctDNA mutation profiling, soluble immune checkpoint molecules, hematologic inflammatory indices, and plasma-based tumor mutational burden (cTMB) represents a promising approach to refine patient selection for immune checkpoint inhibitors (ICIs) and guide treatment adaptation. Such composite biomarker models could help identify not only who is most likely to benefit from immunotherapy but also when to escalate, de-escalate, or discontinue treatment, based on evolving molecular and immune parameters.

Future research should prioritize large-scale, prospective clinical trials designed to validate the predictive and prognostic value of liquid biopsy-derived biomarkers across diverse NSCLC populations and treatment settings. These studies should adopt harmonized protocols for sample collection, assay design, and biomarker interpretation to improve inter-study comparability and clinical utility.

In parallel, the development of advanced computational models, including machine learning and AI-based algorithms, offers an opportunity to synthesize high-dimensional data from multiple biomarker classes. Integrating genomic, transcriptomic, proteomic, and cellular immune profiles into a unified predictive framework could significantly enhance precision in therapeutic decision making. Initial efforts in this space have demonstrated encouraging results, with multimodal models outperforming traditional single-biomarker approaches in predicting ICI response [77].

Furthermore, serial liquid biopsy sampling enables longitudinal monitoring of minimal residual disease (MRD), early detection of resistance mechanisms, and timely identification of relapse. This capacity may ultimately support adaptive treatment strategies and reduce reliance on invasive tissue re-biopsies, particularly in the metastatic setting.

As liquid biopsy technology continues to mature, several key challenges must still be addressed, including regulatory standardization, reimbursement policy, and equitable access to testing. Nevertheless, its clinical promise is undeniable. With continued innovation and rigorous clinical validation, liquid biopsy is poised to become a cornerstone of immunotherapy-guided care in thoracic oncology.

In conclusion, liquid biopsy represents a transformative advancement in NSCLC management. By enabling real-time molecular profiling, immune monitoring, and adaptive treatment decision making, it has the potential to refine the application of ICIs and targeted therapies, minimize toxicity, and ultimately improve clinical outcomes in patients with lung cancer.

## Figures and Tables

**Figure 1 genes-16-00954-f001:**
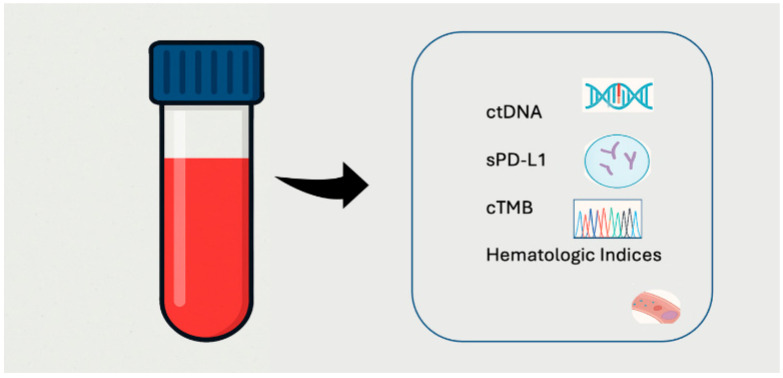
Liquid Biopsy Components Highlighted in This Review: Molecular and Immune Profiling Tools.

**Figure 2 genes-16-00954-f002:**
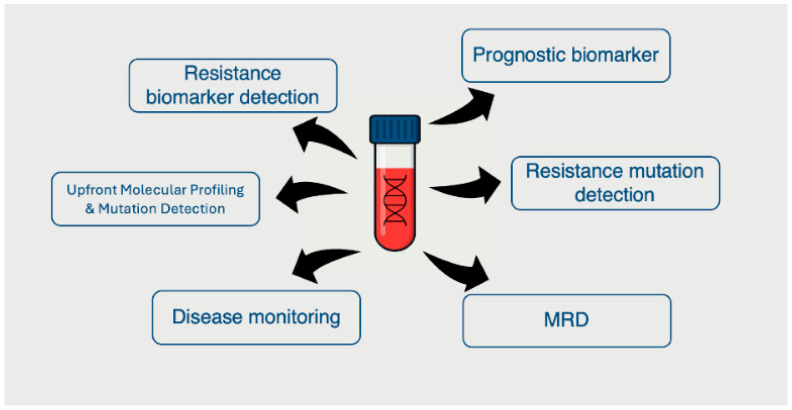
ctDNA as a source of predictive and prognostic biomarkers in NSCLC, enabling mutation profiling, resistance monitoring, minimal residual disease detection, and dynamic assessment of treatment response.

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
