# Peer review of "Beyond the Tissue: Unlocking NSCLC Treatment Potential Through Liquid Biopsy"

_genes, 2025, doi:10.3390/genes16080954_

Round 1

Reviewer 1 Report

Comments and Suggestions for Authors

I have read the article submitted for review with great attention. The study is comprehensive and of high medical quality. NSCLC is an important topic in medicine both from the point of view of the incidence of this cancer and the high percentage of associated mortality.
The perspective of liquid biopsy as a therapeutic potential represents a modern and future approach.

The authors present an extensive review of possible existing biomarkers as well as diagnostic methods.

I noticed some small citation problems. There are paragraphs that refer to statistical data and we do not have the references (lines 36-43; 89-94; 355-360). To improve the article, it is necessary to review all the citations in the article.

Author Response

Dear Editor,

On behalf of all authors, I would like to thank you and the reviewers for the thorough and constructive evaluation of our manuscript. We are pleased that the manuscript will be reconsidered for publication in Genes, and we have carefully revised the text to address all comments.

Below is our detailed, point-by-point response to the reviewers’ remarks. All changes are incorporated in the revised manuscript, with modifications tracked for your convenience.

We look forward to your feedback.

Kind regards,

Assoc. Prof. Milica Kontic Jovanovic, MD, PhD

Reviewer’s comments:

Reviewer 1: I have read the article submitted for review with great attention. The study is comprehensive and of high medical quality. NSCLC is an important topic in medicine both from the point of view of the incidence of this cancer and the high percentage of associated mortality.
The perspective of liquid biopsy as a therapeutic potential represents a modern and future approach.The authors present an extensive review of possible existing biomarkers as well as diagnostic methods.

I noticed some small citation problems. There are paragraphs that refer to statistical data and we do not have the references (lines 36-43; 89-94; 355-360). To improve the article, it is necessary to review all the citations in the article.

We sincerely thank the reviewer for pointing out these omissions. We carefully reviewed the manuscript and added appropriate references to support the statements on:

  • the global burden of lung cancer and the predominance of NSCLC (lines 36–43),
  • recent technological advances enabling sensitive detection of circulating biomarkers (lines 89–94), and
  • the prognostic relevance of ctDNA clearance (lines 355–360).

Additionally, we conducted a comprehensive review of the entire manuscript and ensured that all statistical claims, clinical outcomes, and specific data points are now properly referenced using current and peer-reviewed sources. We believe this improves the scientific rigor and citation quality of the article.

Reviewer 2 Report

Comments and Suggestions for Authors

In this manuscript, Kontic et al. provides a systematic review of the critical role of liquid biopsy in non-small cell lung cancer (NSCLC), particularly in the context of immunotherapy related works. They authors put emphasis on information extracted from circulating tumor DNAs, and also briefly mentioned research on soluble biomarkers and cell type ratios in liquid biopsy. The authors also discussed how these can support personalized treatment in NSCLC and conjugation with novel concepts in clinical and AI research. The review is comprehensive on literature coverage. However, the overall structure of the review needs further improvement for better clarity, and can include some analyses on limitations of cited research.
The structure of a few sections needs significant improvement. For instance, section 3 in connected with subsections of the section 4 without a formal title for the section 4. The organization of paragraphs lacks structural consistency. In some cases, introductory context and supporting literature are presented in the same paragraph, while in others, individual studies with only subtle differences are discussed in separate paragraphs. Although this may be minor, such inconsistency can affect the logical flow and overall readability of the review. Also, the title of the section 3 is not clear and has inconsistent word capitalization.
In addition, although there are several sections focusing on different targets or applications in biopsy, the main targets are mainly ctDNA, small molecules and peptides, or cells. Current organization mixed the targets and the application platforms, which should be improved to better deliver key points to readers.

Author Response

Dear Editor,

On behalf of all authors, I would like to thank you and the reviewers for the thorough and constructive evaluation of our manuscript. We are pleased that the manuscript will be reconsidered for publication in Genes, and we have carefully revised the text to address all comments.

Below is our detailed, point-by-point response to the reviewers’ remarks. All changes are incorporated in the revised manuscript, with modifications tracked for your convenience.

We look forward to your feedback.

Kind regards,

Assoc. Prof. Milica Kontic Jovanovic, MD, PhD

Reviewer’s comments:

Reviewer 2: In this manuscript, Kontic et al. provides a systematic review of the critical role of liquid biopsy in non-small cell lung cancer (NSCLC), particularly in the context of immunotherapy related works. They authors put emphasis on information extracted from circulating tumor DNAs, and also briefly mentioned research on soluble biomarkers and cell type ratios in liquid biopsy. The authors also discussed how these can support personalized treatment in NSCLC and conjugation with novel concepts in clinical and AI research. The review is comprehensive on literature coverage. However, the overall structure of the review needs further improvement for better clarity, and can include some analyses on limitations of cited research.
The structure of a few sections needs significant improvement. For instance, section 3 is connected with subsections of the section 4 without a formal title for the section 4.

We thank the reviewer for noting this. The section numbering and headings have now been corrected. Section 4 has been given a formal title, with its subsections clearly defined below.

The organization of paragraphs lacks structural consistency. In some cases, introductory context and supporting literature are presented in the same paragraph, while in others, individual studies with only subtle differences are discussed in separate paragraphs. Although this may be minor, such inconsistency can affect the logical flow and overall readability of the review.

We thank the reviewer for this valuable observation. To improve structural consistency and enhance the logical flow of the manuscript, we have revised the organization of several paragraphs throughout the text. These modifications were made consistently across the manuscript, following a clear pattern of introducing the concept, summarizing grouped evidence, and providing interpretative context. We believe these changes improve the coherence and readability of the review.

Also, the title of the section 3 is not clear and has inconsistent word capitalization.

We appreciate the reviewer’s comment. The original section title has been revised for clarity and consistency. The new title reads:

“ctDNA Kinetics as a Predictor of Immunotherapy and Targeted Therapy Response”

This updated title corrects the capitalization, removes redundancy, and more accurately reflects the content and focus of the section.

In addition, although there are several sections focusing on different targets or applications in biopsy, the main targets are mainly ctDNA, small molecules and peptides, or cells. Current organization mixed the targets and the application platforms, which should be improved to better deliver key points to readers.

We thank the reviewer for this insightful suggestion. To address this concern, we have clarified the structure of each section to more explicitly distinguish between the type of biomarker (e.g., ctDNA, soluble proteins, immune cells) and its clinical application (e.g., response prediction, resistance detection, disease monitoring). Where appropriate, we added introductory statements at the beginning of relevant sections to define the biomarker type and highlight the specific applications discussed. We believe these refinements help delineate the content more clearly and improve readability and comprehension for the reader.

Reviewer 3 Report

Comments and Suggestions for Authors

The article presents the subject thoroughly with appropriate context for the available data. It includes a clinical perspective on the implications and limitations of liquid biopsy. The topic is relevant in current discussions.

Please review the titles. The number has changed from 3 to 4.1..

Author Response

Dear Editor,

On behalf of all authors, I would like to thank you and the reviewers for the thorough and constructive evaluation of our manuscript. We are pleased that the manuscript will be reconsidered for publication in Genes, and we have carefully revised the text to address all comments.

Below is our detailed, point-by-point response to the reviewers’ remarks. All changes are incorporated in the revised manuscript, with modifications tracked for your convenience.

We look forward to your feedback.

Kind regards,

Assoc. Prof. Milica Kontic Jovanovic, MD, PhD

Reviewer’s comments:

Reviewer 3: The article presents the subject thoroughly with appropriate context for the available data. It includes a clinical perspective on the implications and limitations of liquid biopsy. The topic is relevant in current discussions. Please review the titles. The number has changed from 3 to 4.1.

We thank the reviewer for noting this. The section numbering and headings have now been corrected. Section 4 has been given a formal title, with its subsections clearly defined below.
